# Cooperatively Routing a Truck and Multiple Drones for Target Surveillance

**DOI:** 10.3390/s22082909

**Published:** 2022-04-10

**Authors:** Shuangxi Tian, Xupeng Wen, Bin Wei, Guohua Wu

**Affiliations:** 1College of Systems Engineering, National University of Defense Technology, Changsha 410073, China; tsx_0726@nudt.edu.cn; 2College of Business Administration, Hunan University of Finance and Economics, Changsha 410205, China; 3School of Traffic and Transportation Engineering, Central South University, Changsha 410075, China; a1363904299@gmail.com (B.W.); guohuawu@csu.edu.cn (G.W.)

**Keywords:** target surveillance, truck and drone, two-echelon routing, adaptive large-scale neighborhood search

## Abstract

With the development of drone technology, drones have been deployed in civilian and military fields for target surveillance. As the endurance of drones is limited, large-scale target surveillance missions encounter some challenges. Based on this motivation, we proposed a new target surveillance mode via the cooperation of a truck and multiple drones, which enlarges the range of surveillance. This new mode aims to rationally plan the routes of trucks and drones and minimize the total cost. In this mode, the truck, which carries multiple drones, departs from its base, launches small drones along the way, surveils multiple targets, recycles all drones and returns to the base. When a drone is launched from the truck, it surveils multiple targets and flies back to the truck for recycling, and the energy consumption model of the drone is taken into account. To assist the new problem-solving, we developed a new heuristic method, namely, adaptive simulated annealing with large-scale neighborhoods, to optimize truck and drone routes, where a scoring strategy is designed to dynamically adjust the selection weight of destroy operators and repair operators. Additionally, extensive experiments are conducted on several synthetic cases and one real case. The experimental results show that the proposed algorithm can effectively solve the large-scale target surveillance problem. Furthermore, the proposed cooperation of truck and drone mode brings new ideas and solutions to targets surveillance problems.

## 1. Introduction

### 1.1. Research Motivation

Drones have the advantages of high flexibility, high timeliness, low cost, and freedom from the limitations imposed by human physiological constraints [1]. Therefore, they have been widely used in surveillance missions in civil and military fields, such as agriculture [2,3,4], search and rescue [5,6], aerial photography and surveillance [7], and detection and defense Systems [8]. In addition, drones can accurately obtain high-precision and high-resolution multivariate data from areas of interest. Therefore, they have become important tools for target surveillance missions in harsh and dangerous battlefield environments [9]. In recent studies, the drone target surveillance problem has been oriented to static targets on the ground, and multiple heterogeneous drones have been considered [10]. Compared with a single drone system, the application of multiple drones can result in more surveillance missions and improve the efficiency of surveilling targets [11].

In this study, the target surveillance problem focuses on how to rationally plan the routes of trucks and drones to improve the surveillance efficiency and minimize the total cost. However, the limited endurance of a single drone makes it difficult to surveil large-scale targets distributed over a wide range [12]. Therefore, we study the target surveillance routing problem by cooperatively using a truck and multiple drones to surveil targets. As shown in Figure 1, the truck, which is loaded with multiple drones, starts at the base, surveils its targets and then returns to the base. Drones can take off and land from the truck. After a drone takes off, it can surveil multiple targets and then return to the truck within its maximum flight range. As shown in Figure 1, drone1 takes off at the 3rd target and surveils the 8th and 9th targets. Then, it returns to the truck at the 5th target. The 3rd and 5th targets are surveilled by the truck. By this process, the truck increases the drone’s surveillance range. Besides, the truck has enough energy when conducting the surveillance mission and both trucks and drones can surveil targets.

However, traditional optimization solvers and exact algorithms may not obtain satisfactory solutions in an acceptable time. Therefore, we design a new method to assist in target surveillance problem solving. This method consists of two main parts: (1) Solution initialization: The initial solution of the surveillance problem is generated according to the nearest neighbor and the preservation strategies, and the idea of maximum cost savings is adopted by replacing several truck targets with drone targets. (2) Solution optimization: An adaptive simulated annealing with large-scale neighborhoods (ASALN) algorithm is developed to optimize the surveillance routes of the truck and drones. In ASALN, multiple destroy and repair operators are designed, and the probability of operators is dynamically adjusted by roulette wheel selection. Meanwhile, multiple scoring operators are developed to update the weight of the selection probability.

### 1.2. Research Contribution

The main contributions of this paper are summarized as follows:

(1) A new target surveillance mode based on the cooperation of a truck and multiple drones is studied. In this mode, the truck, which carries multiple drones, starts from the base and returns to the base after finishing the surveillance of all the targets. Both the truck and drones can surveil the targets. Drones take off from the truck, surveil a number of targets, and return to the truck.

(2) A novel heuristic method is proposed to assist in target surveillance problem-solving. In this method, the initial solution is generated according to the nearest neighbor algorithm of the maximum cost savings strategy, and then an adaptive simulated annealing with large-scale neighborhoods algorithm is developed to further optimize the routes of the truck and drones. Specifically, a scoring strategy is designed to dynamically update the selection weights of the destroy and repair operators.

(3) Extensive experiments are conducted on synthetic and real cases, and the experimental results show that the proposed method can effectively solve the multitarget surveillance problem. Furthermore, sensitivity experiments on several crucial parameters of this method are analyzed.

### 1.3. Research Organization

The brief overview of this paper is organized as follows. In Section 2, the relevant works on the targets surveillance problem and related methods are briefly reviewed. In Section 3, the model assumptions and development of the studied targets surveillance problem are presented. Section 4 elaborates the proposed optimization algorithms to assist the problem-solving, Section 5 reports and discusses the experimental results. Section 6 draws the conclusion and provides future research directions.

## 2. Related Work

The target surveillance problem based on drones mainly focuses on targets on the ground, and a corresponding heuristic algorithm is proposed [13]. Huang et al. [14,15] studied the collaboration with public transportation vehicles (PTVs) and the deployment of charging stations for aerial surveillance by UAVs, and proposed an iterative algorithm to optimize the locations of the charging stations. Yang et al. [16] studied the routing problem of multiple drones traveling to multiple targets by using a particle swarm optimization algorithm. Wu et al. [17] proposed a cooperative routing algorithm that meets drone maneuvering constraints and adapts to the data communication delay for the cooperation of multiple drones in uncertain environments. The authors in [18,19] studied spatiotemporal cooperative routing to ensure that multiple drones reach their targets and made attempts to consider communication constraints in the routing of multiple drones. Kuo et al. [20] considered the task time window in the drone task allocation model and proposed the variable neighborhood search procedure with a novel solution representation as a solver. Boso et al. [21] proposed the neural dynamic programming method on the premise that the positions and heading directions of different drones could be shared among the swarm. Hu et al. [22] studied multidrone routing in surveillance missions to ensure full coverage of surveillance areas. Tian et al. [23] established that target surveillance requires multiple cooperative drone surveillance models. Ahn et al. [24] considered a multiobjective vehicle routing problem with drones for military surveillance operations. With the development of drone operational application technology, drones can land on carrier platforms to carry out surveillance missions but are no longer allowed to access a fixed base for scheduling and task planning. Gonzalez et al. [25] solved the truck drone team logistics (TDTL) problem by mixed integer planning, which generates routes the drone must follow to visit all specified locations and assigns rendezvous points where the drone’s batteries are replaced from the truck.

Researchers have been increasingly interested in the cooperation of trucks and drones for performing complex tasks, such as parcel delivery problems, road network surveillance problems and target surveillance problems. However, the target surveillance problem in this study focuses on rationally planning the routing of trucks and drones. In recent years, many routing problems for trucks and drones have been studied, and corresponding algorithms have been proposed. Murray and Chu [26] proposed the flying sidekick traveling salesman problem (FSTSP), in which a truck launches a drone at a delivery point and the drone returns to the truck after delivering to series of customers in a flight. This problem is described as the mixed integer linear programming (MILP) problem. The heuristic algorithm of FSTSP first solves the TSP route as truck routing problem and then decides whether to assign the eligible drone to the drone route or reinsert it to the truck route at a different delivery location. Agatz et al. [27] proposed a similar problem, the traveling salesman problem with drone (TSP-D). Murray and Raj [28] extended the problem in which only single trucks and single drones are considered and studied the problem of the cooperation of trucks and multiple drones. Since then, many other extended problems and improved algorithms have been proposed, including modified FSTSP routing problems [29,30,31,32], extended multitruck and multidrone routing problems [33,34,35], and more effective optimization algorithms [36,37,38,39,40,41,42]. In addition, Poikonen and Golden [43] proposed the mothership and drone routing problem (MDRP), where the drone is allowed to visit multiple targets sequentially before returning to the mothership for refueling, and an exact branch-and-bound and two greedy heuristic methods are proposed to solve the problem.

It can be found that in the previous studies, only trucks or only drones are considered in the target surveillance problem, while the cooperation of trucks and drones has not yet been addressed. Different from the truck-only or drone-only mode, the cooperative routing of trucks and drones for the target surveillance problem is a variant of the two-echelon routing problem (2E-RP). The main distinguishing features of this problem are that both the truck and drones can surveil targets, a drone takes off from the truck and surveils multiple targets in a flight, and flies to the truck for recycling. Multiple drones can simultaneously surveil targets. Eventually, the truck returns to the base with all the drones. As a result, the truck can be used as a mobile carrier to extend the endurance of drones; thus, more surveillance missions can be accomplished, and the surveillance efficiency can be significantly improved. Additionally, an adaptive simulated annealing with large-scale neighborhoods (ASALN) algorithm is designed to effectively solve the target surveillance problem.

## 3. Problem Description and Model Development

### 3.1. Problem Description and Definitions

The target surveillance problem begins with a truck carrying multiple drones to surveil the target set, which is aim at finding the best route for the truck and drones to surveil all targets at the lowest cost while not violating the capacity limit of the drone battery. Essentially, the surveillance problem based on trucks and multiple drones is a two-echelon routing problem (2E-RP), where the first echelon is truck route and the second echelon is drone route. In addition, according to graph theory, each point in the undirected graph represents the distribution center and target location, and the line between two points represents the truck or drone route. Therefore, the undirected graph can be used for the routing problem G=(L,E), where *L*:{$D_{i}$ where *i* = 0,; 1, ;2…, *n*} represents the set of surveillance targets and the base, and $D_{0}$ represents the base, $D_{all}=1,\;2,\;\ldots ,\;n$ represents the set of all surveillance targets, and *E* represents the set of edges in the truck’s (or the drone’s) route.

To clearly express the proposed model, we first list the symbols and definitions used for the cooperation of the truck and drones in the target surveillance problem in Table 1, including the truck and drone parameter definitions, set definitions, and decision variables.

### 3.2. Model Assumptions

The proposed model focuses on studying the target surveillance problem using a truck and multiple drones. The truck carries multiple drones from the base and launches drones at the surveillance target. Then, a small drone surveils multiple targets depending on its battery capacity and flies back to the truck for recycling. The truck surveils targets along its route and finally returns to the base with all the drones. To simplify the problem model, we make the following assumptions:

(1) The truck has enough energy to surveil all the surveillance targets;

(2) Both the truck and drones surveil targets;

(3) Drones only take off/land on the truck when the truck stops at a surveillance target;

(4) The truck must arrive at the landing point before the drone arrives;

(5) The truck surveillance process does not affect the taking off/landing the drones;

(6) The weight of the drone is constant, and the truck and drones maintain a uniform speed;

(7) The unit costs of the truck and drones are fixed. That is, the change in the unit driving cost of the truck caused by the weight change of the truck-mounted drone is not considered.

### 3.3. Energy Consumption Model of the Drone Battery

Since the energy consumption rate is a crucial factor for drone endurance, it is of great significance to estimate the energy consumption model during the surveillance of multiple targets in the drone subroute. For the convenience of presentation, it is assumed that the drone’s self-weight is $w_{d}$, its flight speed is a fixed value $v_{2}$, and its power is *p*. When the type of drone is determined, the values of all relevant parameters of the drone are known. To achieve the highest surveillance efficiency, we assume that the drone flies at the highest speed, i.e., it maintains its maximum power during flight.
(1)$$v_{2}=V$$
(2)p=P
where *P* denotes the maximum power, and *V* denotes the maximum flight speed. Thus, the flight time and energy consumption from the *i*-th surveillance target to the *j*-th surveillance target can be calculated as follows:(3)$$t_{ij}=\frac{{L_{ij}}^{\prime }}{V}$$
(4)$$F_{ij}=Pt_{ij}=P\frac{{L_{ij}}^{\prime }}{V}$$

On this basis, the total energy $F_{ij}$ consumed by the drone in a subroute can be calculated to determine whether it exceeds the capacity of the battery. Meanwhile, the cost of drones in a subroute can be further calculated.

### 3.4. Mathematical Formulation

To understand the definition of the problem more accurately, a mathematical model is established for truck and drone surveillance missions. Given the network G=(L,E), each truck route starts from the base $D_{0}$, visits multiple surveillance targets in $D_{all}$ and then returns to base $D_{0}$, as shown in Figure 1, i.e., the sequence of the route is (0, 1, 2, 3, 4, 5, 6, 0). Each route (r∈R) should satisfy two constraints: The first constraint is that the in-degree and out-degree of any target in *r* (including the surveillance targets and base) must be equal to 1; the second constraint is that there is no subloop in *r*. For any directed main route by truck r(r∈R), its cost can be calculated and denoted as $c_{r}$.

Suppose the truck route is r(r∈R), where $D_{r}$ is the set of surveillance targets visited by the truck; then, all other surveillance targets in $D_{r}\prime$ should be surveilled by the drones. Meanwhile, assuming that all route segments *l* in the given main route r(r∈R) can be enumerated, let $S_{r}$ be the set of all route segments in the main route *r*. For route segment *l* in the main route $r(l\in S_{r})$, assume that the feasible subroutes corresponding to the route segment are denoted as $R_{rl}$. Each drone’s subroute $m(m\in R_{rl})$ starts from the first surveillance target in the route segment, visits several surveillance targets $D_{r}\prime$, and ends at surveillance target *l*. Suppose Drlm denotes the surveillance target set in drone subroute $m(m\in R_{rl})$, and $E_{rlm}$ denotes the edge set in drone subroute $m(m\in R_{rl})$.

According to the energy consumption model, the drone flight cost of subroute m can be calculated and expressed as crlm. Suppose $x_{r}\in$ {0,1}, $y_{rlm}\in$ {0,1}, $a_{ijrlm}\in$ {0,1}, $b_{irlm}\in$ {0,1}, and $c_{ir}\in$ {0,1} are binary variables. If the truck chooses the main route r(r∈R), then $x_{r}$=1; otherwise, $x_{r}$ = 0. Given the truck route (r∈R), the directed subroute $l\in S_{r}$. If the drone chooses subroute $m(m\in R_{rl})$, then $y_{rlm}$ = 1; otherwise, $y_{rlm}$ = 0. Given two surveillance targets $i,j(i,j\in D_{all})$ and a drone subroute $m(m\in R_{rl})$, if edge $\left(i,j\right)\in E_{rlm}$, then $a_{ijrlm}$ = 1; otherwise, $a_{ijrlm}$ = 0. If $i\in D_{rlm}$, then $b_{irlm}$ = 1; otherwise, $b_{irlm}$ = 0. If $i\in D_{r}$, then $c_{ir}$ = 1; otherwise, $c_{ir}$ = 0.

Based on the above assumptions and definitions, we establish the 2RP-T&D mathematical formulation based on the two-echelon route problem. In the first echelon, a heuristic algorithm is proposed to optimize the drone subroute on the premise of a given truck route, which can be used to calculate the lowest cost of all main routes *R*. In the second echelon, the overall cost of the truck and drones is iteratively optimized.

Model 1 (Drone subroute optimization): For any given truck route r(r∈R) and definitely set $S_{r},D_{r}$, and $D_{r}\prime$, the feasible drone subroutes corresponding to the main route *r* can be calculated and denoted as $R_{rl}$. Then, the minimum cost of the drone’s subroute accessing all targets in $D_{r}\prime$ can be calculated by Model 1, which can be expressed as follows:(5)$$Minz_{r}=\sum_{l,m}c_{rlm}y_{rlm}$$
(6)$$s.t.\sum_{l,m}a_{ijrlm}y_{rlm}\leq 1,\;\forall i,j\in D_{r}$$
(7)$$\sum_{l,m}b_{irlm}y_{rlm}=1,\;\forall i\in D{}_{r}^{\prime }$$
(8)$$\begin{array}{c} \sum_{i\in D_{r},j\in {D_{r}}^{\prime }}a_{ijrlm}=\sum_{i\in {D_{r}}^{\prime },j\in D_{r}}a_{ijrlm}=n\\ \forall m\in R_{rl},\forall l\in S_{r} \end{array}$$
(9)$$\begin{array}{c} at_{i}{}^{\prime }+\left(d{t_{i}}^{\prime }+\frac{L_{ij}{}^{\prime }}{v_{2}}\right)a_{ijrlm}-M\left(1-a_{ijrlm}\right)\leq a{t_{j}}^{\prime }\\ \forall i,j\in D_{all},i\neq j,\forall m\in R_{rl},\forall l\in S_{r} \end{array}$$
(10)$$\sum_{\left(i,j\right)\in E_{rlm}}F_{ij}\leq W,\;\forall m\in R_{rl},\forall l\in S_{r}$$
(11)$$y_{rlm}\in \left\{0,1\right\},\;\forall m\in R_{rl},\;\forall l\in S_{r}$$
(12)$$\sum_{l}y_{rlm}\leq N_{d},\;\forall m\in R_{rl}.$$

Objective function (5) minimizes the cost of the drone subroute [44]. For a given main route r(r∈R), constraint (6) ensures that each truck route segment l corresponds to at most one subroute of a drone to prevent the crossing of drone routes. Constraint (7) forces each drone surveillance target to be visited only once. Constraint (8) imposes the takeoff or landing of drones at only truck surveillance targets. Constraint (9) is the time window constraint of drones, which ensures that drones can only conduct surveillance operations only after reaching the surveillance target. Constraint (10) restricts the energy constraint of the drone. Constraint (11) defines the decision variables. Constraint (12) refers to the number constraint of vehicle-mounted drones.

Model 2 (Truck main route optimization): For the truck main route optimization, the lowest total cost is obtained. The lowest subroute cost corresponding to truck main route *r* denotes $z_{r*}$, which can be obtained through Model 1. The truck main route optimization model is shown as follows:(13)$$MinZ_{}=\sum_{r}c_{r}x_{r}+\sum_{r}x_{r}{z_{r}}^{*}$$
(14)$$s.t.\sum_{r}x_{r}^{}=1$$
(15)$$\sum_{r\in R}c_{ir}x_{r}=1\forall i\in D_{r}$$
(16)$$\begin{array}{c} at_{i}+\left(dt_{i}+\frac{{L_{ij}}^{}}{v_{1}}\right)c_{ir}-M\left(1-c_{ir}\right)\leq at_{j}\\ \forall i,j\in D_{all}\cup D_{0},i\neq j,\forall r\in R \end{array}$$
(17)$$\begin{array}{c} at_{i^{\prime }}\prime +\left(dt_{i}+\frac{L_{ij}}{v_{1}}\right)c_{ir}-M\left(1-c_{ir}\right)-M\left(1-c_{jr}\right)-\\ M\left(1-a_{i^{\prime }jrlm}\right)\leq at_{i^{\prime }}{}^{\prime }+\left(d{t_{i^{\prime }}}^{\prime }+\frac{L_{i^{\prime }j}{}^{\prime }}{v_{2}}\right)a_{i^{\prime }jrlm}\\ \forall i,i^{\prime },j\in D_{all}\cup D_{0},i\neq j,i^{\prime }\neq j,\\ \forall m\in R_{rl},\forall l\in S_{r},\forall r\in R \end{array}$$
(18)$$\begin{array}{c} at_{i^{\prime }}{}^{\prime }+\left(d{t_{i^{\prime }}}^{\prime }+\frac{L_{i^{\prime }j}{}^{\prime }}{v_{2}}\right)a_{i^{\prime }jrlm}-M\left(1-c_{ir}\right)-M\left(1-c_{jr}\right)\\ -M\left(1-a_{i^{\prime }jrlm}\right)\leq at_{i}+\left(dt_{i}+\frac{L_{ij}}{v_{1}}\right)c_{ir}+te_{i} \end{array}$$
(19)$$\begin{array}{c} \forall i,i^{\prime },j\in D_{all}\cup D_{0},i\neq j,i^{\prime }\neq j,\\ \forall m\in R_{rl},\forall l\in S_{r},\forall r\in R,x_{r}\in \left\{0,1\right\},\forall r\in R \end{array}$$

Objective function (13) minimizes the total cost, where $z_{r*}$ is obtained by Model 1. Constraint (14) ensures that only one truck main route can be selected. Constraint (15) forces each truck surveillance target to be visited only once. Constraints (16–18) denote the time window constraints. Constraint (16) restricts that the surveillance can be conducted only after the truck arrives; Constraints (17–18) ensure that the truck reaches the landing point before the drone and leaves the landing point after the drone lands. Constraint (19) defines the decision variables. For the directed main route r(r∈R), note that there may be no feasible solution in Model 1, in which case, $z_{r*}$ is set as infinity *∞*.

Based on the above model, the surveillance problem based on the 2RP-T&D model can be optimized in two main steps. First, the main route of the truck is obtained so that the set R can be obtained. For the main route r(r∈R), the subroutes of the drones are obtained by Model 1. Second, the optimized route of the truck and drones is obtained by Model 2.

## 4. Algorithm Design

With the increase in the scale of the target surveillance problem using a truck and multiple drones, it is difficult to obtain high-quality solutions in an acceptable time using an exact algorithm or optimization solver. Therefore, we design a method to assist problem solving in this section, where the nearest neighbor and cost savings strategies are first developed to obtain the initial solution. Then, the adaptive simulated annealing with large-scale neighborhoods algorithm is developed to optimize the initial solution, in which a scoring strategy is designed to dynamically update the selection weights of the destroy and repair operators. The flowchart of the proposed method is presented in Figure 2.

### 4.1. Nearest Neighbor Cost-Savings Strategy (NNCS)

Inspired by the idea of truck first drone second, we design a heuristic algorithm combining nearest neighbor and cost savings strategy (NNCS). This strategy is graphically represented in Figure 3. First, the nearest neighbor search is applied to construct the main route of the truck, which is shown in Figure 3a. Then, the maximum cost savings are adopted by replacing several trucks with drones, which is shown in Figure 3b. The pseudocode of the NNCS algorithm is presented in Algorithm 1.

From Algorithm 1, the current surveil target d′, the target set that never surveilled Du, the target set that has already been surveilled $D_{s}$, the truck route $S_{0}$ and the drone route $S_{1}$ are initialized (line 1). First, the main route of truck is constructed (lines 3–8). The target $d_{c}$ nearest to d′ from $D_{u}$ is found, and it is reassigned to the current target d′ (line 3). The target $d_{c}$ is removed from the unsurveil target set $D_{u}$, and the target $d_{c}$ is added to the surveilled target set $D_{s}$ (lines 4–6). When the truck returns to base, a truck-only route $S_{0}$ is generated (line 8).

Then, several truck targets are replaced with drone targets according to the order of the surveillance targets in the main route (lines 10–18). In each iteration, the cost-savings $C_{si}$ (line 11) for surveillance target *i* is calculated, and the surveillance target with the maximum cost-savings Csimax is found (line 12). If $C_{simax}\geq 0$, target *i* is removed from truck route $S_{0}$, and target *i* is added to drone route $S_{1}$ (lines 13–15). If $C_{simax}<0$, then the loop is terminated (lines 16–17), and the initialized solution $S=\{S_{0},S_{1}\}$ is output (line 18).
**Algorithm 1:**NNCS($D_{all},D_{0}$)
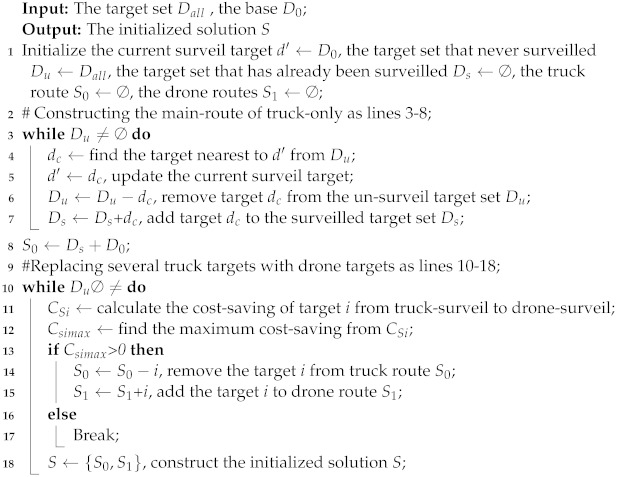


### 4.2. Adaptive Simulated Annealing with Large-Scale Neighborhoods (ASALN)

In this section, the proposed ASALN algorithm is presented. The destroy operators and repair operators designed in ASALN are described, and the adaptive adjustment strategy developed in ASALN is presented to dynamically select the destroy and repair operators. In addition, simulated annealing is introduced as an iterative optimization framework, and the tabu list is utilized to avoid short-term loops and make the routing algorithm for trucks and drones more effective.

#### 4.2.1. Design of Destroy and Repair Operators

Since adaptive large-scale neighborhoods have significant advantages, the self-definition of the operator, which increases the diversity of solutions, is easy to expand, and the added heuristic information improves the algorithm search efficiency [40], which is suitable for solving the routing problem of the truck and drones. However, when the size of the neighborhoods gradually increase, the time complexity increases exponentially. To improve search efficiency, the proposed ASALN algorithm allows multiple neighborhoods to be searched in a single search, the neighborhood operator weight is adjusted according to the solution’s quality, and the neighborhood of the next iteration is selected based on its weight. Specifically, the design of the destroy and repair operators is elaborated as follows.

(1) Destroy Operator Design

When the NNCS algorithm obtains an initial solution, the destroy operator is generated by roulette wheel selection and *q* surveillance targets are removed from the route of the truck or the drones, resulting in a route $S_{p}$ that is missing *q* targets and has *q* deleted targets. In subsequent repair operations, the *q* surveillance targets will be reinserted into $S_{p}$.

Considering that the drone must take off or land from the surveillance target visited by the truck, the destroy operation is divided into two types, as shown in Figure 4a. If the selected target does not include the takeoff or landing point of the drone, it can be deleted directly. If the selected target consists of the takeoff or landing point of the drone, as shown in Figure 4b, a new takeoff or landing point needs to be constructed when the drone takeoff or landing point is deleted. Specifically, if the drone takeoff point is deleted, the previous point is taken as the new drone takeoff point; if the drone landing point is deleted, the next point is taken as the new landing point.

There are two specific destroy operator designs, i.e., random destroy operators and maximum saving destroy operators.

##### Random Destroy Operators

The random destroy operator randomly removes the *q* surveillance target from the route of the truck and the drones, resulting in a partial solution $S_{p}$ that is missing *q* surveillance targets and has *q* removed surveillance targets, as shown in Figure 5. Random destroy operators enhance the randomness of heuristic algorithms and thus increase the diversity of solutions.

##### Maximum Savings Destroy Operators

The maximum savings destroy operators remove the *q* surveillance targets with the highest cost increase in the route of the truck and drones. Thus, a partial solution $S_{p}$ is missing *q* surveillance targets and has *q* removed surveillance targets, as shown in Figure 6. For solution *s*, the cost savings of removing target *i* are defined as $cost\left(i,s\right)=f\left(s\right)-f_{i}\left(s\right)$, where f(s) is the cost of the current solution and $f_{i}\left(s\right)$ is the cost of the solution after removing surveillance target *i*. The cost savings for each surveillance target are calculated and recorded, and the surveillance target with the largest cost savings in the existing solution is removed. Then, the current solution is updated, the cost savings of all surveillance targets are recalculated, and the surveillance targets with the largest cost savings are deleted. This process is repeated until *q* surveillance targets are deleted. The maximum saving destruction operator makes the algorithm obtain a smaller cost portion of the solution $S_{p}$, and the surveillance targets have more costs removed. The reconstruction of the route may cause the targets to be inserted into a different location than the current solution, which will reduce the cost of the surveillance problem.

(2) Repair Operator Design

When *q* surveillance targets are deleted by the destroy operator, a repair operator is executed based on roulette wheel selection. According to the repair operator, the algorithm will insert *q* surveillance targets and reinsert them into the route of the truck or drone to obtain a new solution $S^{\prime }$. Specifically, two repair operators are designed, the greedy repair operator and the regret repair operator.

##### Greedy Repair Operator

The greedy repair operator inserts *q* deleted surveillance targets into the partial solution $S_{p}$ with a greedy strategy; thus, a new complete solution $S^{\prime }$ is obtained, which is graphically presented in Figure 7. The greedy repair operator first inserts each of the *q* surveillance targets previously deleted into the truck and drone routes of partial solution $S_{p}$ and selects the surveillance targets and locations corresponding to the lowest cost solution. Then, the insertion action is executed to obtain the new partial solution $S_{p}^{\prime }$ and the remaining q−1 surveillance targets. The insertion cost of the remaining surveillance targets in the new solution $S_{p}^{\prime }$ is recalculated to determine the next surveillance targets to be inserted. Then, the *q* deleted surveillance targets are successively added to the partial solution $S_{p}$.

##### Regret Repair Operator

Different from the greedy repair operator, the regret repair operator first selects an appropriate surveillance target according to the extra cost and then inserts it into the appropriate position to minimize the total cost. The regret value insertion operator determines the insertion of *q* deleted surveillance targets according to the regret value of the insertion cost.

The regret value is the cost difference between the best insertion position and the rest insertion position of surveillance targets in the route. The higher the regret value is, the more significant the cost difference between the optimal insertion position and the suboptimal insertion position of the surveillance target. Therefore, surveillance targets with high regret values should be inserted into the truck and drone routes. The regret value is calculated as follows:(20)$$\max_{i\in U}c_{i}{*}^{}=\Delta {f_{i}}^{2}-\Delta {f_{i}}^{1}$$
where $\Delta f_{i1}$ denotes the increased cost after inserting surveillance target *i* at the optimal insertion location, $\Delta f_{i2}$ denotes the increased cost after inserting surveillance target *i* at the suboptimal location, $c_{i*}$ denotes the regret values of surveillance target *i* inserted at different locations, and *U* is the set of deleted surveillance targets.

According to the characteristics of the surveillance problem, it is necessary to calculate the regret value of the surveillance target inserted into the two different routes of truck and drone and compare the regret value of all the different insertion targets of each surveillance target in the truck and drone routes to obtain the maximum value of the regret value of each surveillance target.

First, the increased cost of each previously deleted surveillance target *i* after the insertion of each feasible insertion point is calculated, and the increased cost of the surveillance target *i* after the insertion of the optimal insertion point is subtracted to obtain the maximum regret value of each surveillance target. Then, the largest regret of each surveillance target is compared, and the maximum regret value is selected. Thus, the surveillance target that needs to be inserted and its location can be determined, and then the regret value of the deleted surveillance targets is recalculated. The *q* previously deleted surveillance targets are added to the partial solution $S_{p}$ in iteration until all the surveillance targets are inserted into the truck and drone routes.

#### 4.2.2. Adaptive Adjustment Strategy (AAS)

For the selection strategy of destruction and repair, an adaptive adjustment strategy (AAS) is designed. The selection weight $w_{j}$ of the adaptive adjustment operator is determined according to the number of destroy and repair operators used.

Specifically, the core idea of the designed adaptive strategy is to calculate the score of destroy and repair operators by three bonuses, i.e., the higher the score is, the better the performance of the operator. The first bonus $\sigma_{1}$ occurs when generating new and improved solutions by deleting and inserting operations. The second bonus $\sigma_{2}$ occurs when accepting an unaccepted solution that is better than the current solution. The third bonus $\sigma_{3}$ occurs when accepting an unaccepted solution that is worse than the current solution.

The process of added bonuses is divided into several periods. In each period, the score of the operators is initialized to zero. The roulette probability of all operators is the same. After each additional period, the total score of each operator in the previous period is used to calculate the new weight, and the formula for calculating the weight is presented as follows:(21)$$\omega_{i,j+1}=\omega_{i,j}\left(1-r\right)+\frac{r\pi_{i}^{}}{\theta_{i}}$$
where $\omega_{ij}$ denotes the weight of operator *i* in period *j*, $\pi_{i}$ denotes the scores of operator *i*, and $\theta_{i}$ denotes the number of uses of operator *i*. Influence factor *r* controls the change speed and change proportion of weight. The algorithm determines the probability that the next periodic operator will be selected by roulette according to the new weight. The better the operator performs, the higher its score and weight, and the greater the probability that the operator will be used. The pseudocode of the adaptive adjustment strategy is presented In Algorithm 2.
**Algorithm 2:**AAS ($Z_{best}$, *Z*, Z′, $\pi_{i1}$, $\pi_{i2}$, $\sigma_{1}$, $\sigma_{2}$, $\sigma_{3}$)
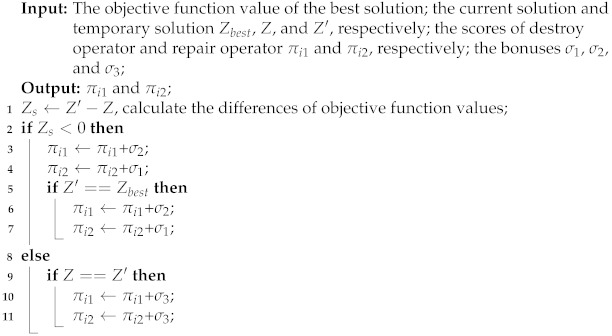


The input of Algorithm 2 is the objective function values of the best solution, current solution and temporary solution, which are $Z_{best}$, *Z*, and Z′, respectively. First, the differences in objective function $Z_{s}$ are calculated. If the temporary solution is better than the current solution, then the operator score is the score plus $\sigma_{2}$ (lines 2–4). If the best solution is obtained, then the operator score is the score plus $\sigma_{1}$ (lines 5–7). If accepting a temporary solution that is worse than the current one, then the operator score is the score plus $\sigma_{3}$ (lines 9–11). Thus, the operator scores $\pi_{i1}$ and $\pi_{i2}$ are updated and finally output.

#### 4.2.3. The Proposed ASALN Algorithm

As mentioned above, the proposed ASALN algorithm adopts the SA algorithm as the iterative optimization framework. Specifically, the core idea of SA is to accept the poor solution with a certain probability that may jump out the local optima and may obtain the global optimum. In addition, the lack of short-term memory easily leads to short-term loops and revisiting targets. Therefore, the efficiency of SA can be improved by memorizing the tabu list. In the search for a target surveillance problem, the new solution is accepted if it costs less than the old one, or the new solution is accepted with a certain probability if the new solution costs more than the old one. Once the new solution is accepted, the selected neighborhood is added to the tabu list. Without loss of generality, neighbor solutions in the tabu list cannot be inserted into the new solution until the temperature drops. In all, the pseudocode of the adaptive simulated annealing with large-scale neighborhoods is presented in Algorithm 3.
**Algorithm 3:**The proposed ASALN
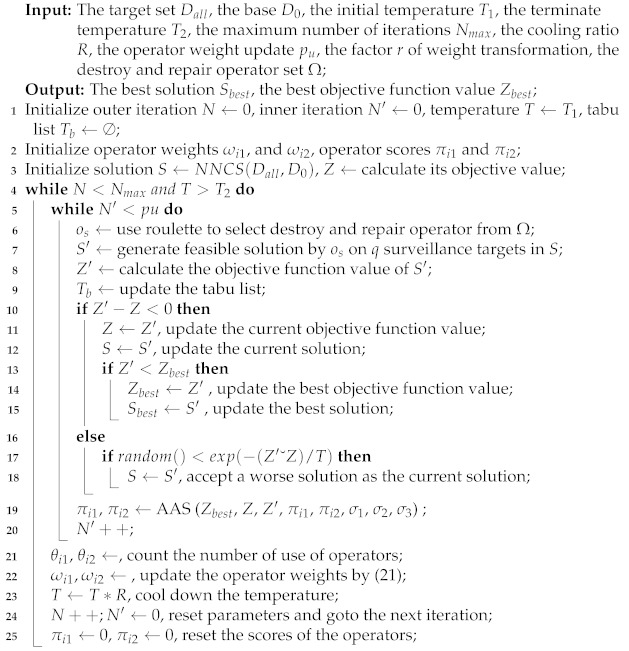


From Algorithm 3, the input is target set *D*, the initial temperature $T_{1}$, the terminate temperature $T_{2}$, the maximum number of iterations $N_{max}$, cooling ratio *R*. First, parameters of iterations and the tabu list are initialized (line 1), and the operator weights $\omega_{i1}$ and $\omega_{i2}$, the operator scores $\pi_{i1}$ and $\pi_{i2}$ are initialized (line 2). The initial solution is obtained by the NNCS algorithm (line 3). At each temperature, the number of iterations and tabu list is set to improve the performance of the inner loop (line 5). In each iteration, the algorithm adopts a roulette strategy to select destroy and repair operators and the current neighborhood is generated (line 6). The temporary solution is obtained from the current neighborhood S′ (line 7), and its objective function value Z′ is calculated (lines 8). The invalid moves are added to the tabu list (line 9). If the difference (Z′–Z) is less than 0 (line 10), the solution S′ is considered the best solution (line 12). If the temporary solution is better than the best solution, the best solution is reset to the temporary solution (lines 13–15). In addition, a poor solution may be accepted with a certain probability according to the criterion of random()<exp(−(Z′˘Z)/T) (lines 17–18). At the end of each iteration, the operator scores are updated by the AAS algorithm (line 19). After the inner loop, the number of uses of the destroy operator $\theta_{i1}$ and repair operators $\theta_{i2}$ are counted (line 21), the weights of the destroy and repair operators are updated (line 22), the temperature is cooled down (line 23), and the scores of the operators are reset to zero in the next iteration (lines 24–25). When the termination condition of SA is met, the best solution $S_{best}$ and the best objective function value $Z_{best}$ are output.

## 5. Experiments and Analysis

In this section, three scales of the synthetic cases are generated to verify the performance of the proposed algorithm. In addition, a real case of target surveillance, i.e., 100 traffic surveillance targets in Changsha, China, is conducted for comparison with different surveillance modes. Furthermore, the sensitivity of crucial factors of the truck and drone for surveillance problems is analyzed, including the scale of the drone surveillance target in the initial solution, the influence of the number of drones, and the battery power of drones.

### 5.1. Experimental Design and Settings

The environment is conducted on an Intel Core i5-8300H CPU with 8 GB memory. The experimental cases include several synthetic cases and a real case. The synthetic cases include small-scale, medium-scale, and large-scale cases, and the real case is selected from traffic surveillance targets in Changsha.

(1) Surveillance case setting

The surveillance targets of synthetic cases are generated in the same way as in [40]. The surveillance area is divided into four parts, and each part generates the same number of surveillance targets. For the synthetic cases, each of the small, medium, and large surveillance scales generates ten instances. The number of surveillance targets and the surveillance area dimensions corresponding to the target surveillance scale are listed in detail in Table 2. For the real case, 100 traffic intersection surveillance targets in Changsha are selected according to the longitude and latitude of traffic intersections, which are obtained from Baidu and converted into the relative coordinate values x and y according to the Miller coordinate system.

(2) Algorithm parameter settings

To verify the effectiveness of the proposed algorithm, we compare it with a truck-only surveillance algorithm and three other cooperation of truck and drone surveillance algorithms, including the SA algorithm, the TS algorithm and the truck-only (TO) algorithm. The parameter settings of these algorithms are as follows.

ASALN algorithm: According to the settings in [45], the iteration $p_{u}$ is set to 5 times to ensure the timeliness of the operator weight update. The influence factor *r* of weight transformation is set as 0.4. The bonus $\sigma_{1}$ for deleting and inserting operations that generate new and better solutions is 33. Bonus $\sigma_{2}$ for accepting an unaccepted solution that is better than the current one is 9. Bonus $\sigma_{3}$ for accepting an unaccepted solution that is worse than the current one is 13. In addition, the initial temperature $T_{1}=\omega *c_{i}/log\left(2\right)$, where ω = 0.05, $c_{i}$ denotes the cost of the initial solution; the initial temperature $T_{1}$ is 6000 ${}^{{}^{\circ}}$C; the terminating temperature $T_{2}$ is 100 ${}^{{}^{\circ}}$C; the cooling ratio *R* is 0.9; and the maximum number of iterations $N_{max}$ is the same as $p_{u}$ and is set to 5. The above parameters are set according to the settings in [45].

SA algorithm: The parameters in the SA search are the same as those in the SA-TS search, $T_{1}=\omega *c_{i}/log\left(2\right)$. The cooling function $T_{k}=T_{s}*0.99k$ is adopted according to the settings in [46]; the initial temperature $T_{1}$ is set to 6000, and the terminate temperature $T_{2}=T_{1}/50$. The cooling ratio *R* is set to 0.99.

TS algorithm: According to the settings in [47], the maximum number of iterations and the maximum number of unimproved solutions are set to 100 and 10, respectively.

TO algorithm: According to the settings in [48], the initial temperature $T_{1}$ is set to 93${}^{{}^{\circ}}$C, the termination temperature $T_{2}$ is set to 3 ${}^{{}^{\circ}}$C, and the cooling ratio *R* is set to 0.99.

(3) Truck and drone settings

The parameters of the truck and drones are set according to the parameters commonly used in surveillance problems. The truck can travel up to 400 km with a full tank of gas. Therefore, it can complete surveillance missions without refueling. In addition, truck costs include energy consumption and operating expenses. According to the settings in [44], a delivery truck has an efficiency of six miles per gallon when using diesel and consumes 0.392 L per kilometer, and the cost of a truck is $0.717/km. In addition, the operating cost of the truck is $0.484/km after factoring in maintenance, depreciation and driver wages. Thus, the total truck operating cost is $1.201/km.

According to the parameters of the Amazon octa-rotor drone [49], the weight of the drone is set to 2 kg, and the weight of the surveillance camera loaded by the drone is set to 2 kg. Based on the energy consumption model, the drone has a power of 5000 mAh and a maximum sailing distance of 14 km. For the cost of the drone, according to the work in [44], it can be estimated that the unit energy cost of the drone is $1.345 ×$10^{-4}$/mAh. When the battery power of the drone is 5000 mAh and the maximum range is 14 km, the energy consumption cost per unit distance of the drone is approximately $0.048/km. After considering the drone cost and depreciation fee, the cost of a drone is approximately $0.45/km. Thus, the total cost of drones is approximately $0.498/km. Therefore, the parameters of the truck and drone can be determined as shown in Table 3.

### 5.2. Experimental Analysis on Synthetic Cases

On each scale, five examples are generated using the above experimental settings and parameters. For each case, the initial solution obtained by the NNCS is compared with the solution obtained by ASALN. Meanwhile, the solutions generated by the proposed ASALN algorithm are compared with those generated by the traditional SA and TS algorithms. Each case runs ten times, and the average is then calculated. The experimental results on synthetic cases at three scales are shown in Table 4.

Comparison with the NNCS algorithm: From Table 4, for the small-scale cases, compared to the cost of the initial solution obtained by the NNCS, the cost of the solution obtained by the proposed ASALN algorithm is reduced by 38.78% to 45.90%. For the medium-scale cases, compared to the cost of the initial solution obtained by the NNCS, the cost of the ASALN algorithm is reduced by 22.64% to 42.81%. For the large-scale cases, the cost of the ASALN algorithm is reduced by 33.40% to 42.81%. In most cases, the final cost calculated by the ASALN algorithm is more than 30% lower than the cost calculated by the NNCS, which proves the effectiveness of the proposed ASALN algorithm.

Comparison with SA and TS algorithm: The experimental results of the SA and TS algorithms on the small-scale, medium-scale, and large-scale cases are recorded in Table 4. It can be found that the cost of the ASALN algorithm is similar to the cost of the SA algorithm. However, with the increase in the case scale, the SA algorithm is more time-consuming, and takes up to 1.27 times longer to run than the ASALN algorithm. The reason can be inferred as the lack of heuristic information in the SA algorithm which easily leads to short-term repetition. Compared with the TS algorithm, the experimental results of the ASALN algorithm are more stable, and they become sensitive to the increase in the case scale. On the large-scale cases, the ASALN algorithm has a cost savings that is 7.90% higher (25.21% total) compared to the TS algorithm. However, in terms of algorithm computational time, the computation time of the ASALN algorithm is longer than that of the TS algorithm, but it can still be controlled within 2 min. Considering the cost savings, the computation time of the ASALN algorithm can be accepted.

Comparison with truck-only surveillance mode: Two different surveillance modes, i.e., the cooperation of truck and drone target surveillance ASALN and truck-only (TO) target surveillance, are compared. From Table 4, it can be observed that the mode of cooperation of trucks and drones can dramatically reduce the total cost compared to that of the truck-only mode. With the assistance of drones, the final cost generated by the ASALN is more than 70% lower than that of the truck-only model. The comparative analysis of the two modes is shown in Figure 8.

### 5.3. Experimental Analysis on a Real Case

(1) Source of data: One-hundred surveillance target sites are selected in Changsha, China. Specifically, the latitude and longitude of the locations in this experiment are obtained from Google Maps and converted into Miller coordinates.

(2) Analysis of experimental results: The surveillance route of the truck and drones is shown in Figure 9, where the bold line represents the truck route, and the fine lines of different colors represent the routes of the drones. From Figure 9, 29 targets are surveilled by the truck and 71 targets are surveilled by the drones. A total of six drones are needed. In addition, from the route shown in Figure 9, the main route of the truck is relatively smooth, while remote surveillance targets and the surveillance targets that may cause round-trip trends are surveilled by drones, which significantly reduces the total cost.

Figure 10 presents the convergence curve of the proposed ASALN algorithm. The termination conditions of ASALN adopts the termination criteria of simulated annealing, where the temperature would drop at each iteration until the termination temperature is reached. Specifically, the parameters are set as follow: the initial temperature $T_{1}$ is set to 6000 ${}^{{}^{\circ}}$C, the terminating temperature $T_{2}$ is set to 100 ${}^{{}^{\circ}}$C, and the cooling ratio *R* is set to 0.9. From Figure 10, the objective function value converges at the 64th iteration, and the number of iterations is within an acceptable range. In the first 20 iterations, the objective function value converges quickly, and the operator’s reconstruction of the initial solution is more intense. In the iterative process, the convergence curve can continuously find a promising solution. The reason can be speculated to be that the ASALN algorithm allows multiple neighborhoods to be searched in a single search, which increases the diversity of solutions. Meanwhile, the SA algorithm can jump out of the local optimum, and the TS algorithm can avoid reducing the repeated search of the solution space of poor quality to improve the search efficiency of the ASALN method.

### 5.4. Sensitivity Analysis

Based on the above real case, sensitivity analyses are conducted on three crucial factors, i.e., the proportion of drone surveillance targets in the initial solution, the number of vehicle-mounted drones, and the drone endurance.

(A) Sensitivity analysis of drone surveillance target proportion in the initial solution: To investigate the influence of drone surveillance target proportion in the initial solution, the proportions of drone surveillance targets are set as 25%, 35%, 45%, 55% and 65% of the total number of targets. The experimental results are shown in Table 5.

From Table 5, it can be found that the proportion of drone surveillance targets in the initial solution has little influence on the objective function value when other parameters remain the same. In the initial solution, when drone surveillance targets are set to 65% of the total surveillance targets, “—” means that the number of vehicle-mounted drones is insufficient to complete the surveillance of all drone surveillance targets.

In addition, the influence of the drone surveillance target scale on algorithm convergence in the initial solution is shown in Figure 11. By observing the convergence of different curves, it can be seen that the curves can converge quickly in 100 generations. It can be inferred that this is the optimization effect of the proposed ASALN algorithm on the initial solution. Although the proportions of drone surveillance targets in the initial solution are different, the initial solution is destroyed and repaired by the proposed ASALN algorithm, which makes the proportion of drone surveillance targets and truck surveillance targets in the final solution tend to be the same. Therefore, the above convergence curves of four scales can converge well and there is little difference between the convergent values. However, the different scales of drones in the initial solution lead to different convergence rates of the proposed ASALN algorithm. Specifically, as the proportions of drone surveillance targets go from 25% to 45%, the scale of the drone surveillance target in the initial solution gradually enlarged and the convergence rate becomes faster. When the proportion of drone surveillance targets in the initial solution reaches 55%, the convergence rate of the proposed ASALN algorithm becomes slowed down. When the proportion of drone surveillance targets in the initial solution is 65%, there is no feasible solution due to the limited number of drones.

(B) Sensitivity analysis of vehicle-mounted drone number: Since vehicle-mounted drones can only be used once in a surveillance mission, the number of vehicle-mounted drones determines the upper limit of the number of drone surveillance routes. To investigate the influence of the number of vehicle-mounted drones, different numbers of drones are set under the condition that other parameters are unchanged. The experimental results are shown in Figure 12.

From Figure 12, as the number of vehicle-mounted drones increases, the targets in the best solution can make cost-savings after changing their surveillance mode, and the targets can be surveilled by the drones, thus saving cost. Therefore, it is necessary to explore the appropriate number of vehicle-mounted drones for a specific case. As the number of vehicle-mounted drones increases from 3 to 9, the average number of targets surveilled by drones is decreased, and some targets that are originally surveilled by trucks become surveilled by drones. Combining the results in Table 4, the cost of surveilling targets by the truck decreases, while the cost of surveilling targets by drones increases, and the total cost decreases. Specifically, the total cost decreased from $20.52 to $17.74, a reduction of 13.6%. Truck costs decreased from $15.78 to $8.73, a reduction of 44.7%. Drone costs increased from $4.74 to $9.01, an increase of 90%. By observing the curve trend in Figure 12, it can be seen that with the increase in the number of vehicle-mounted drones, the rate of total cost reduction gradually slows down. When there are more than 6 drones, the total cost remains constant. In this case, the drones appear idle, and blindly increasing drones may produce more costs. The endurance of drones becomes the main constraint. In addition, as the operators in the ASALN algorithm always give priority to the target that saves the most costs, the rate of cost-savings should be expressed as high to low, which is consistent with the results in Figure 12.

(C) Sensitivity analysis of drone endurance: Drone endurance, as an important parameter of a drone, determines the maximum number of surveillance targets in a flight, and the reasonable selection of different drone endurances is of great help to save costs. To investigate the influence of the drone endurance on total cost, the drone endurance is set to 10–22 km, and the experimental results are shown in Figure 13.

From Figure 13, as the drone endurance increases, some targets surveilled by the truck are changed to be surveilled by drones, which reduces the cost of the truck and increases the cost of the drones, meanwhile the total cost decreased from $21.14 to $17.34, a reduction of 18%. Truck costs decreased from $13.92 to $8.73, a reduction of 37.3%. Drone costs increased from $4.74 to $9.01, an increase of 19.4%. The reason can be explained as follows. In the process of optimization solutions, adding a truck target to an existing drone route may yield cost-savings, while the drone may not afford surveillance missions at that target due to endurance constraints. However, adding that target to a new drone route would increase the total cost. Increasing the endurance of the drone would generate a better solution. In addition, when the drone has a range of more than 16 km, the total cost remains constant. In this case, there are no extra suitable targets for drone surveillance, and the drone has surplus energy. The targets outside the drone surveillance route are closer to the truck’s route; thus, the cost of surveillance by the truck becomes smaller, and the remaining energy is always few so as to the second take-off will not be enough to complete the surveillance mission or can only surveil fewer targets, and this part of the energy is often wasted.

## 6. Conclusions

In this paper, we study the target surveillance problem based on a new mode, i.e., the cooperation of trucks and drones, This new mode focuses on how to rationally plan the routes of trucks and drones to improve the surveillance efficiency and the aim of this target surveillance problem is to minimize the total cost. In this mode, a truck departs from the base and surveils targets along the way and returns to the base with all drones. After being launched from the truck, drones surveil multiple targets and then return to the truck. Based on this motivation, we propose a method ASALN, where the nearest neighbor and cost-saving strategy is designed to generate initial solutions to the target surveillance problem. Then, an adaptive large-scale neighborhood strategy is developed under the simulated annealing framework to optimize the routing of the truck and drones. In addition, extensive experiments are conducted on real and synthetic cases, and the results show that the proposed method ASALN has competitive performance than the SA and TS algorithms on the small-scale, medium-scale and large-scale cases of multitarget surveillance problems. Therefore, the proposed cooperation of truck and drone mode brings new ways to targets surveillance problem.

In the future, we will further explore the target surveillance problem via the cooperation of multiple trucks and multiple drones. Additionally, the objective function of this paper is to consider the minimum cost of the target surveillance problem are not comprehensive enough, studing the customer satisfaction as an objective function is meaningful work. Therefore, in the next stage of research work, further studying the multi-objective optimization problem to improve the level of multi-target surveillance problems is a promissing direction, where considering the objective of minimum cost and the objective of customer satisfaction simultaneously.

This research was funded by the National Natural Science Foundation of China under Grant 62073341.

## Figures and Tables

**Figure 1 sensors-22-02909-f001:**
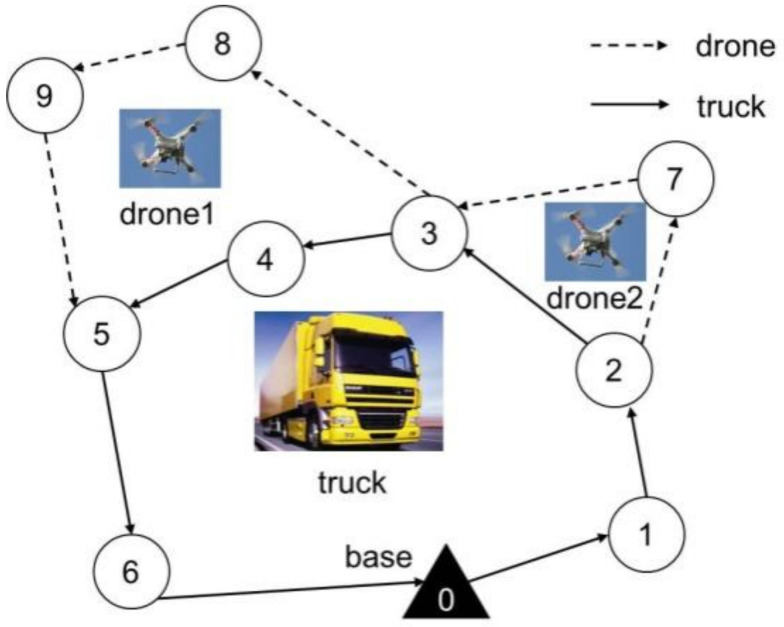
A truck and multiple drones for target surveillance.

**Figure 2 sensors-22-02909-f002:**
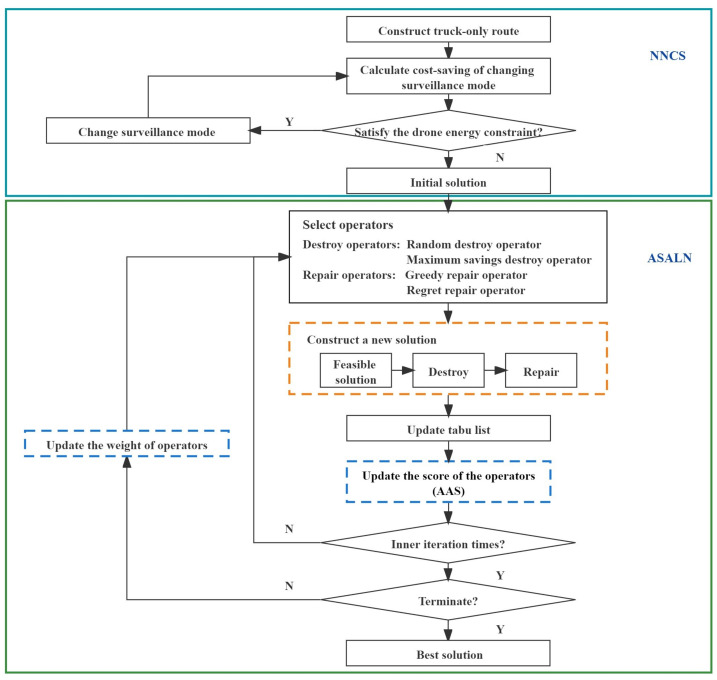
The flowchart of the proposed method.

**Figure 3 sensors-22-02909-f003:**
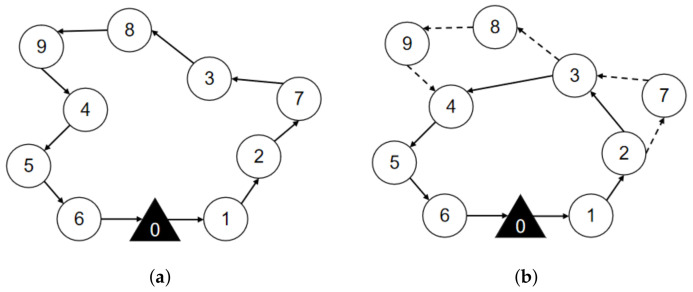
The initial solution schematic diagram by combining the nearest neighbor and cost-savings strategies. (**a**) Truck only for surveillance targets. (**b**) Cooperation of a truck and drones for surveillance targets.

**Figure 4 sensors-22-02909-f004:**
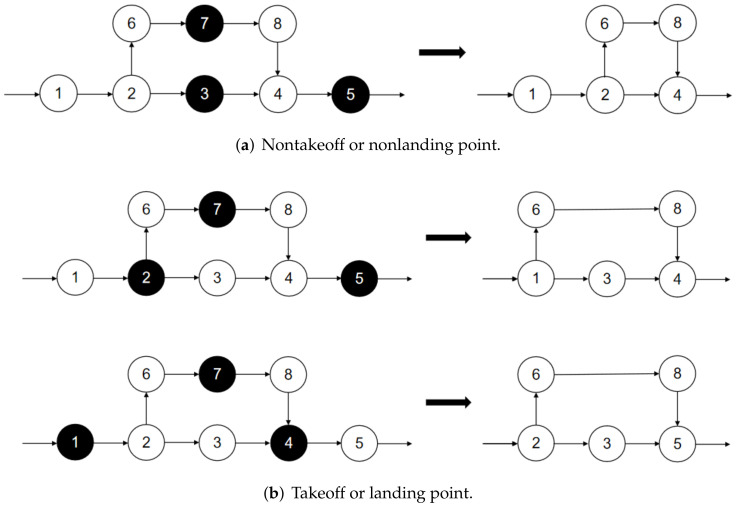
Schematic diagram of destroy operators.

**Figure 5 sensors-22-02909-f005:**
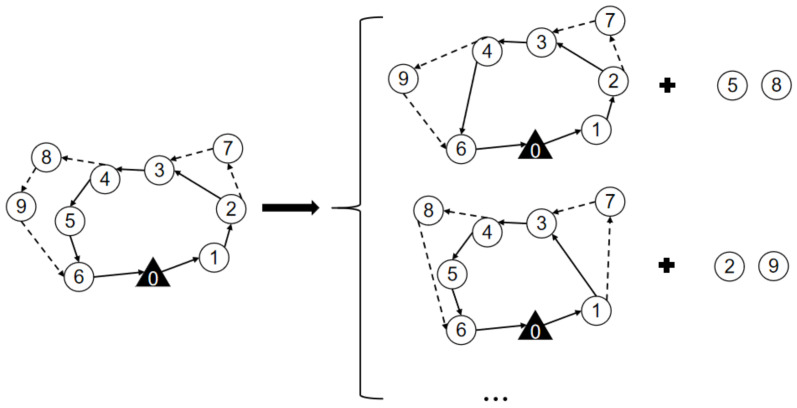
Schematic diagram of the random destroy operator when *q* = 2.

**Figure 6 sensors-22-02909-f006:**
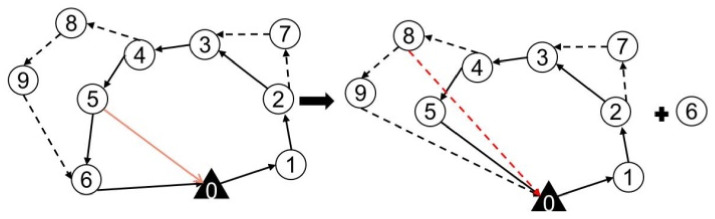
Schematic diagram of the maximum savings destroy operator.

**Figure 7 sensors-22-02909-f007:**
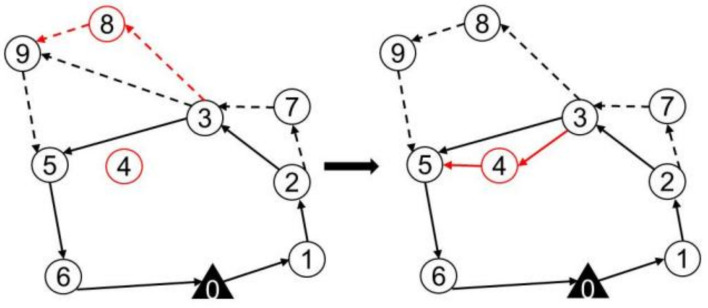
Schematic diagram of the greedy repair operator.

**Figure 8 sensors-22-02909-f008:**
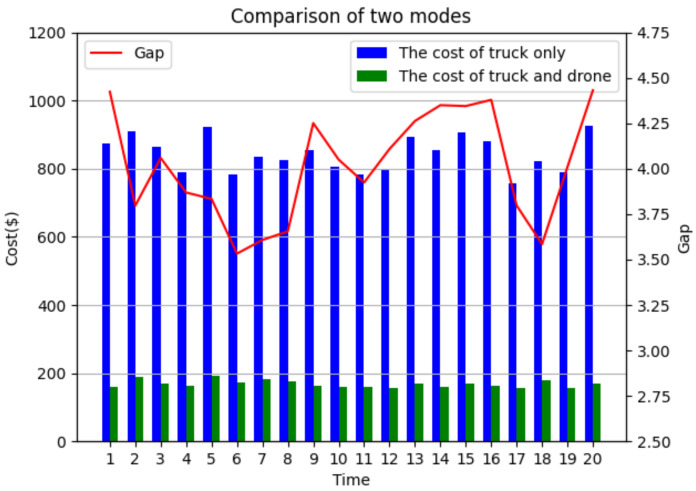
The comparison results of the two surveillance modes.

**Figure 9 sensors-22-02909-f009:**
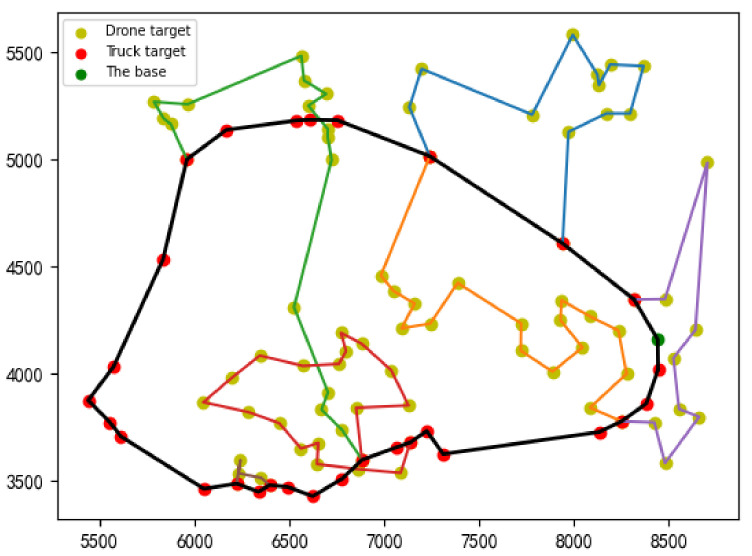
The surveillance route of truck and drone.

**Figure 10 sensors-22-02909-f010:**
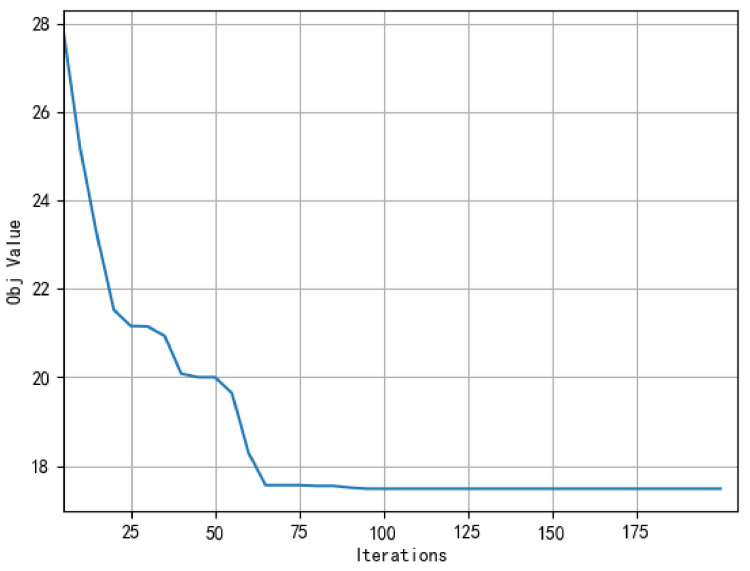
Convergence curve of the objective function value.

**Figure 11 sensors-22-02909-f011:**
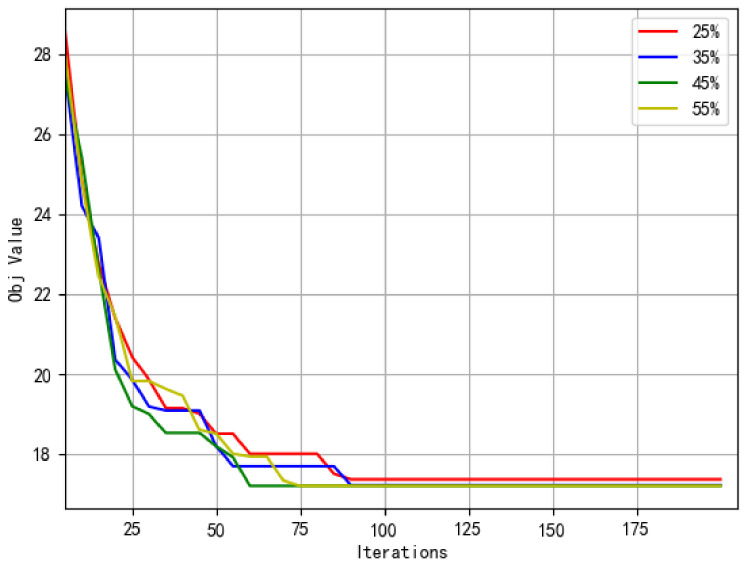
The convergence of different drone surveillance targets in the initial solution.

**Figure 12 sensors-22-02909-f012:**
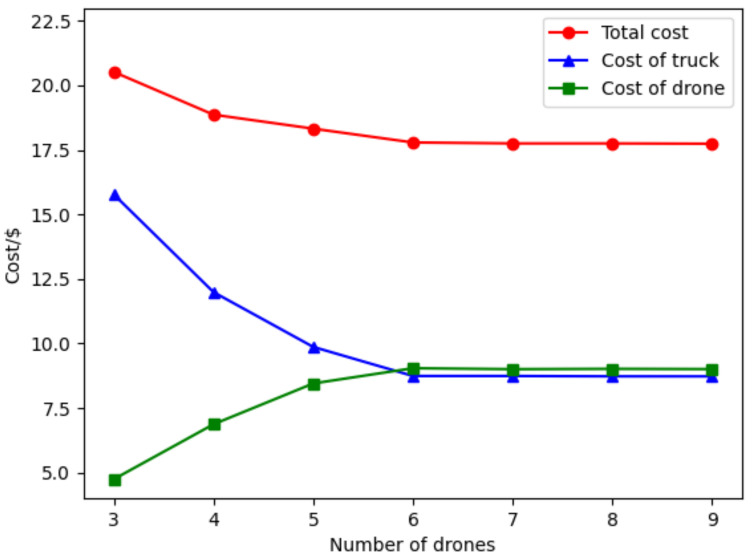
Experimental results under different numbers of drones.

**Figure 13 sensors-22-02909-f013:**
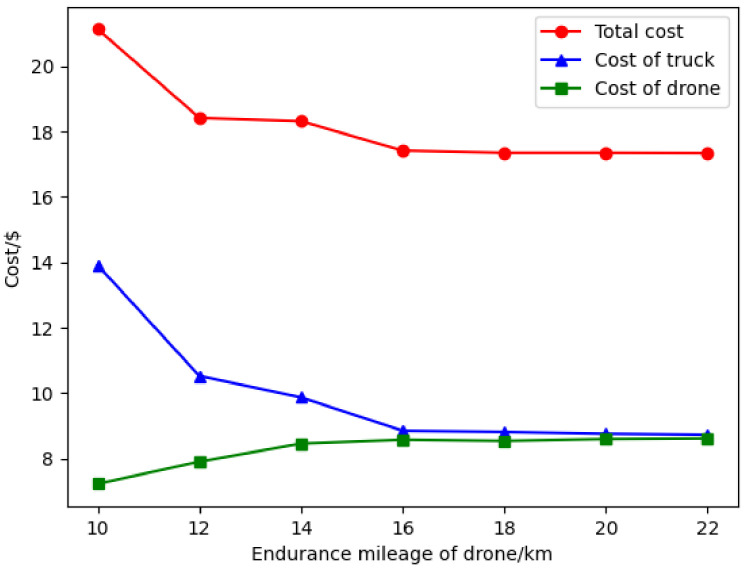
Experimental results under different drone endurances.

**Table 1 sensors-22-02909-t001:** Symbols and definitions.

Symbols	Definitions
*Sets*:	
*G*	The undirected graph of the surveillance problem;
*L*	The set of surveillance targets and the base;
*E*	The set of edges in the truck’s (or the drone’s) route;
$D_{all}$	=1, 2,…, *n*, the set of all surveillance targets;
$D_{0}$	=0, the base of the truck and drones;
*R*	The main route of the truck;
$S_{r}$	The set of route segments contained in truck route r,r∈R;
$D_{r}$	The set of truck surveillance targets in truck route r,r∈R;
$D_{r^{\prime }}$	The set of drone surveillance targets in truck route r,r∈R;
$R_{rl}$	The set of drone routes in truck route r,r∈R;
$D_{rlm}$	The set of surveillance targets in the drone route *m* in route segment *l* in truck route $r,m\in R_{rl},l\in S_{r},rinR$;
$E_{rlm}$	The set of edges in the drone route *m* in route segment *l* in truck route $r,m\in R_{rl},l\in S_{r},rinR$;
*Parameters*:	
*n*	The number of drone take-off points;
$L_{ij}$	The distance from surveillance target *i* to surveillance target *j* in the truck’s route, $i,j\in D_{all}$;
$L_{ij^{\prime }}$	The distance from surveillance target *i* to surveillance target *j* in the truck’s route, $i,j\in D_{all}$;
$s_{1}$	The cost per kilometer of the truck;
$s_{2}$	The flying cost per KWH of the drone;
$w_{d}$	The weight of the drone;
$v_{1}$	The speed of the truck;
$v_{2}$	The flying speed of the drone;
*p*	The power of the drone;
*V*	The maximum flight speed of the drone;
*P*	The maximum power of the drone;
*W*	The maximum battery capacity of the drone;
$t_{ij}$	The time consumption of the drone from *i* to *j*;
$F_{ij}$	The energy consumption of the drone from *i* to *j*;
$T_{M}$	The maximum endurance of the drone;
$at_{i}$	The start time of truck surveillance target *i*;
$dt_{i}$	The time required for the truck at surveillance target *i*;
$te_{i}$	The extra stop time for the truck at surveillance target *i*;
$at_{i}^{\prime }$	The start time of drone surveillance target *i*;
$dt_{i}^{\prime }$	The time required for the drone at surveillance target *i*;
$c_{ij}^{T}$	The cost for the truck to go from target i to target $j,i,j\in D_{all}$;
$c_{r}$	The total cost of the trucks’ main route r,r∈R;
$c_{rlm}$	The flight cost of drone in subroute m corresponding to route segment l of trucks’ main route $r,m\in R_{rl},l\in S_{r},r\in R$;
*M*	The infinite positive numbers;
$N_{d}$	The number of truck-mounted drones;
$\pi_{i1,i2}$	The scores of the destroy and the repair operator;
$\omega_{i1,i2}$	The weights of the destroy and the repair operator;
*Decisions*:	
$a_{ijlm}$	If edge $\left(i,j\right)\in E_{rlm}$, then $a_{ijrlm}$ = 1; otherwise, $a_{ijrlm}=0,i,j\in D_{all},r\in R,l\in S_{r},m\in R_{rl}$;
$b_{ijlm}$	If $i\in D_{rlm}$, then $b_{ijlm}$=1; otherwise, $b_{ijlm}=0,r\in R,l\in S_{r},m\in R_{rl}$;
$c_{ir}$	If $i\in D_{r}$, then $c_{ir}=1$; otherwise, $c_{ir}=0,r\in R$;
$x_{r}$	If the truck chooses the main route *r*, then $x_{r}$ = 1; otherwise $x_{r}=0,r\in R$;
$y_{rlm}$	If the drone chooses the subroute *m* corresponding to the route segment l in the main route *r* of the truck, then $y_{rlm}$ = 1. Otherwise, $y_{rlm}=0,m\in R_{rl},l\in S_{r},r\in R$;

**Table 2 sensors-22-02909-t002:** The setting of synthetic cases.

Scale	The Number of Targets	Surveillance Area Size
Small	50	10 km × 10 km
Medium	80	15 km × 15 km
Large	100	20 km × 20 km

**Table 3 sensors-22-02909-t003:** The parameter settings of the truck and drones.

Truck	Cost	$1.201/km
Drone	Self-weight	2 kg
	Battery power	5000 mAh
	Maximum range	14 km
	Cost	$0.498/km

**Table 4 sensors-22-02909-t004:** Experimental results on synthetic cases on three scales.

	TO	T&D	T&D (TS)		T&D (SA)		T&D (ASALN)		Comparison (%)	
Area Size	Cost ($)	Cost ($)	Cost ($)	Time (s)	Cost ($)	Time (s)	Cost ($)	Time (s)	NNCS (%)	TO (%)
Small Scale	179.89	79.94	44.94	51.32	44.73	63.18	44.73	51.64	44.05	75.13
	197.62	87.26	56.45	51.23	48.52	70.98	49.68	49.88	43.07	74.86
	178.63	78.36	43.87	52.00	42.36	63.65	42.39	52.68	45.90	76.27
	187.35	80.32	55.68	48.64	45.89	60.54	46.30	50.01	42.36	75.29
	202.69	87.68	60.87	50.98	52.93	66.58	53.68	49.68	38.78	73.52
Medium Scale	560.96	170.08	124.87	213.81	105.38	386.63	100.74	275.67	40.77	82.04
	598.67	176.56	136.9	240.39	106.96	356.29	100.98	282.69	42.81	83.13
	585.62	179.65	138.62	252.95	121.94	297.63	121.75	263.95	32.23	79.21
	541.35	180.63	139.75	257.65	120.98	340.95	115.64	289.64	35.98	78.64
	574.98	184.65	157.62	275.65	140.96	360.89	142.85	298.79	22.64	75.16
Large Scale	872.89	241.67	174.75	428.69	167.04	576.52	160.95	489.64	33.40	81.56
	909.65	305.08	217.62	452.97	190.97	562.98	189.64	492.63	37.84	79.15
	863.98	298.63	201.68	421.63	179.00	591.39	170.78	465.19	42.81	80.23
	790.65	278.17	198.65	399.67	162.66	567.99	162.39	458.66	41.62	79.46
	921.78	298.66	254.97	401.03	189.99	577.08	190.69	508.96	36.15	79.31

**Table 5 sensors-22-02909-t005:** Different proportions of drone surveillance targets in the initial solution.

Proportions	25%	35%	45%	55%	65%
Objective values	17.98	17.84	17.72	17.72	—

## Data Availability

Not applicable.

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
