# Peer review of "Cooperatively Routing a Truck and Multiple Drones for Target Surveillance"

_sensors, 2022, doi:10.3390/s22082909_

Round 1

Reviewer 1 Report

General comments:

The paper presents a good literature review, is nicely structured and well written. However, there are some areas where this paper needs improvement (mentioned below) to become more useful for researchers and practitioners.

Specific Comments:

  1. P-2, L-136-137: Notations in paper need improvement. In particular,
    1. “where L = D0 U Dall represents the set of surveillance targets and the base” may be represented as “where L:{Di where i=0,1,2…n} represents the base and set of surveillance targets”.
    2. “…D0 = 0 represents…” should be “…D0 represents…”
  2. L-145-146: “Then, a small drone surveils multiple targets depending on its battery capacity and flies back to the truck for recycling.” This theme is expressed multiple times in paper.
  3. It is not clear in the formulation in section 3 how the redundancy between truck and drone coverage is avoided, so that a target is not surveilled by both drone and truck but at least by either drone or truck.
  4. In figure 7, the bar graph legend is missing. It is not clear which one represents truck only and truck and drone surveillance.
  5. Similarly in figure 8 legend is missing.
  6. L532-541: Fig. 9: what is/are the termination criteria and what was the value of the termination criteria parameter when convergence is achieved? It needs to be explained in the text.
  7. Table 5: The obj function value for 65% drone coverage case is missing. Similarly, convergence curve is missing for 65% case in Fig. 10.

Reviewer 2 Report

1- the novelty of the paper is unclear. This should be indicated in the abstract.

2- there are several assumptions in this research, which make it unrealistic for real-life applications. The authors should justify why they used such assumptions.

3- the energy consumption model of the drones should be justified. Is it a standard energy model? How did you calculate the energy consumed? How the reliability of the model was analysed?

4- this is unclear how the drones paths and the truck routes are aggregated. Why the authors used the nearest neighbor algorithm and cost savings strategy? What is the contribution of this approach to this field of research? Is the cost function the novelty of this approach? There are several works in this area, focusing on the nearest neighbors.

5- for the experimental design, how the authors used the number of targets and field size? What does it show? Have they considered the scalability (small, medium and large) of the proposed approach? Have they repeated the experiments to achieve a specific level of confidence?  

6- again, how do they set the parameters for the truck and drones?

7- the evaluation plan is unclear. How the authors select the evaluation parameters? What about the benchmarks? Is there any comparison to show the benefits of such the proposed approach?

8- what are the key findings of this research? the conclusion should summarises them, and outline the potential further works.  

Reviewer 3 Report

Reviewers’ comments

The authors have written on ‘Cooperatively Routing a Truck and Multiple Drones for Target Surveillance”. I believe the readers would find the manuscript more interesting if most of the sentences in the introduction were appropriately linked to each other. I suggest the service of a language editor to ensure that the manuscript is free from any form of grammatical error. In addition to previously stated observations, the following suggestions should help improve the quality of the manuscript.

  1. The Abstract is well written, but the summary of the methodology is weak. It needs to be re-written. It will include the aim, problem statement, methodology, and contribution of this review paper to the growing body of knowledge.
  2. In the first five lines of the introduction section, the authors should cite the following articles, where they make use of drone technology for their research: DOIs: 10.3390/s21134417, 10.3390/s22041453, and 10.1109/icABCD49160.2020.9183810

3. The introduction needs to be re-written; for example, the authors should create two subsections under the introduction section indicating ‘motivation of the research or research contribution’ and ‘research organization; giving a brief overview of how the sections in the manuscript are divided’ this is for ease of readability.

4. A flowchart should be included just at the beginning of the ‘Materials and Methods Section’ explaining the methodology used in this research.

5. The sensitivity analysis in subsection 5.4 is still not comprehensive enough, and the authors need to explain with clarity how they conducted the sensitivity analysis.

6. The Conclusion section needs to be re-written. The conclusion is supposed to be a section based on the results of this study, and there is also a need to state the implication of this research and how it affects the field of drone technology or any future research. The authors still state their aim, objectives, and discussion in the conclusion section. This section needs to be rewritten.

7. The authors should include a limitation and future recommendation section immediately after the conclusion.

8. Finally, the authors must proofread this manuscript and remove all grammatical errors. And the formatting of the research needs to be redone, so many word fonts in one document.

Round 2

Reviewer 2 Report

The authors addressed my comment. I recommend this paper for publication. 

Author Response

Dear Reviewer:

Thank you for your valuable comments.

Sincerely,
Xupeng Wen

Reviewer 3 Report

Dear Authors,

I am satisfied with your response to the reviewer's comments.

Author Response

(The authors gave the same response as above.)
